# communications
# engineering

# Lab2Field transfer of a robotic raspberry harvester enabled by a soft sensorized physical twin

Kai Junge [1][✉], Catarina Pires[1,2] & Josie Hughes[1]

Robotic fruit harvesting requires dexterity to handle delicate crops and development relying upon field testing possible only during the harvesting season. Here we focus on raspberry crops, and explore how the research methodology of harvesting robots can be accelerated through soft robotic technologies. We propose and demonstrate a physical twin of the harvesting environment: a sensorized physical simulator of a raspberry plant with tunable properties, used to train a robotic harvester in the laboratory regardless of season. The sensors on the twin allow for direct comparison with human demonstrations, used to tune the robot controllers. In early field demonstrations, an 80% harvesting success rate was achieved without any modifications on the lab trained robot.

[1] CREATE Lab, EPFL, Lausanne, Switzerland. [2] Instituto Superior Técnico, Lisbon, Portugal. [✉]email: kai.junge@epfl.ch

Developing robots to contribute to agriculture and food production is of critical importance, and can contribute to many of the UN's Sustainability Development Goals[1,2]. However, agriculture, and in particular harvesting, is a challenging domain for robots. Whilst advanced machinery has enabled the harvesting of many crops from barley to beetroot, many crops have thus far escaped automation[3]. This includes those that are delicate, fragile or have highly complex surrounding environments such as berries, leafy vegetables, or grapes[4], and also those which do not ripen homogeneously. In the last decade, harvesting robots have seen notable developments driven by improvements in the underlying enabling technologies. For instance, reports of commercialized harvesting robots have increased from very few to none[5] to approximately 20 or so cases[6,7]. In particular strawberry picking have seen multiple commercialization success through the use of suction, compressed air blowing, and stem cutting methods[7]. However, more generally, despite over 30 years of research, harvesting robots have shown limited performance improvement[8]. Compared to humans their speed is low, the cost of each device is high, and the enabling technologies are not yet mature[6,8]. It is also a domain where the rate of development of robotic solutions is vastly out-stripped by demand. Through the growing world population[9], alongside the challenges in sustaining the agricultural work force[10](made increasingly apparent through events such as the COVID-19 pandemic[11] and Brexit[12]), there is a real need to develop new methods for harvesting research, to accelerate the development of robotic solutions. Improvements in robotic harvesters could have multi-faceted impact on agriculture[13,14], including a reduction in waste, improvement of produce quality, more stable food security and a reduced environmental impact[5,8,15].

The challenges in developing agricultural robots can be divided in two levels. Firstly the implementation of different robotic technologies poses a practical challenge. Integration of grippers, tactile sensors, navigation, visual localization and classification, and more, which must operate robustly requires time and expertise to achieve[16]. Secondly, the environmental conditions linked to agricultural settings poses large challenges. Outdoor environments can be variable, uncertain, and harsh. This is coupled with every crop, breed and specific instance also being subject to variability[17]. One aspect of the agricultural setting which magnifies the aforementioned challenges is the need for field tests for development and evaluation. This is extremely limiting. Crops are only ripe and ready for harvest for a very short period of time and each harvesting experiment is impossible to re-run or repeat as the specific plant and conditions are constantly changing. To accelerate the design, development and evaluation of harvesting robots, we need to remove the reliance on inefficient and costly field trials and leverage alternative methodologies to meet the escalating food needs of our growing population.

Whilst there have been notable examples of robotic harvesting technologies for crops including strawberries[18,19], sweet peppers[20] and apples[21], examples remain limited[22,23]. Within these, repeated periods of field trials for early stage evaluation or training are commonly reported. For example, in developing robots for lettuce harvesting[24], varying mechanical solutions were evaluated in the field before extensive and repeated field trials were performed to develop the visual and mechanical components of the robotic solution. Similarly, a strawberry harvesting robot[19] has been developed that exploits passive and active elements in a custom gripper showing exciting advances in the mechanical technologies, yet success field deployment relied on field tuning, testing, and evaluation[25]. This reliance on field trials is a bottleneck which limits the development of new technologies and approaches. Given the adept nature of human harvesters,

leveraging their expertise could provide a means of accelerating harvesting robots.

Simulation provides one means of reducing the need for real world experiments and has been successfully applied to a number of robotics domains[26] including locomotion[27], swimming[28] and also manipulation[29]. The application of simulation to agricultural robotics has been shown to aid in the path planning and picking of apples[30], however simulating contact with delicate fruit is more challenging. Using a data-driven approach to generate a virtual environment, a harvesting robot for sweet-peppers has also been used to optimize the planning and fruit detection, however, physical contact with the fruit was also not considered[31]. Simulation has been utilized to aid the design of the force feedback control of a soft gripper[32], however, this was limited to pick and place with the gripper. Another challenge in simulation is how to use rich examples of human task performance to assist in developing a manipulation solution. Although this can be achieved with real2sim2real transfer, it introduces further modeling challenges and is hard to achieve at the level of physical interactions. Thus, although simulation has a clear role to play in planning and design optimization, currently sim2real for the physical harvesting interactions is challenging to achieve[33].

We propose a methodology to accelerate harvesting research by fundamentally changing the means by which harvesting robots are designed, optimized and evaluated. We will specifically focus on raspberry harvesting, a delicate crop to harvesting, and one where over 50% of the costs are attributed to harvesting. Our approach leverages soft robotic technologies to develop a sensorized physical twin. This soft robotic device is designed and tuned to mimic the physical interactions and properties of a raspberry and the plant when harvested[34]. By integrating soft sensors into the physical twin we are able to measure the forces applied to the fruit when harvesting. Fig. 1 illustrates the concept (shown in Fig. 1a–c) and the specific implementation for raspberry harvesting (shown in Fig. 1d–f).

From human demonstration of successful harvest on the sensorized physical twin, we can obtain a target performance for the robot and insights into designing a controller. Using this physical twin, we can then perform lab based trials where we seek to imitate the human across a range of different ripeness settings. Through the embedded sensors on the physical twin, a quantitative metric can be designed to tune control parameters through gradient based cost functions. We propose that by closing the reality gap between the physical twin and the field based plants in the tactile and visual domain, the controller and vision systems from a harvesting robot can be fully trained in the lab, such that they can be translated to the field with minimal or no additional tuning required.

To demonstrate this approach, our physical raspberry twin was used to develop the harvesting and the visual servoing controller for a raspberry harvesting robot. When deployed to the field following only lab based training we record an 80% successful harvesting rate for raspberries with none or minimal damage observed. The vision system also successfully transfers to allow the robot to identify the raspberry and locate within 6.5 mm from the center point of the fruit. To the authors knowledge, we believe this to be first robotic system that has been entirely trained in lab conditions and enabled immediate and successful operation in the field.

## Results

In this section we outline the results obtained from the methodology of using a physical twin to develop a raspberry harvesting robot. The core and novel hardware introduced in this method, the physical twin, is a real world simulation of the field

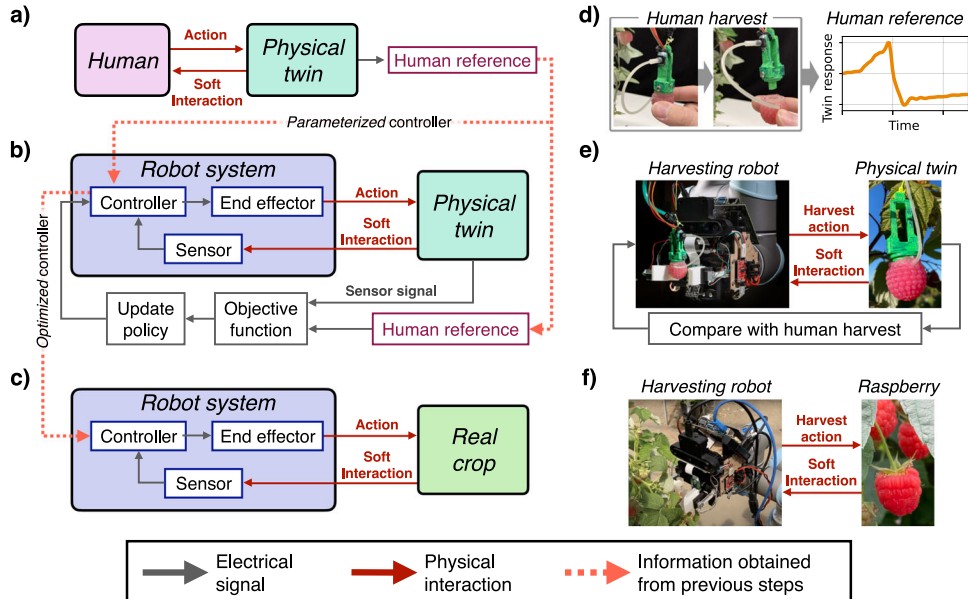

**Fig. 1 The concept and implementation of using a sensorized physical twin for the development and Lab2Field transfer of a raspberry harvesting robot.**
**a**–**c** Describes the concept while **d**–**f** describe the specific implementation of developing a raspberry harvesting robot using a physical twin. **a** The human performs the action we wish the robot to perform on the physical twin to obtain a reference signal and enough insight to design a controller. **b** Using the designed controller, the robot is tested against the physical twin. The human reference is used to tune the controller parameters. **c** The tuned controller is used to deploy the robot into the field to perform the desired action on the real crop. **d** Realization of (**a**) using the physical twin of the raspberry. The human harvests the fruit. **e** Realization of (**b**), the harvesting robot is developed and tuned using the physical twin. **f** Realization of (**c**), the harvesting robot is deployed in the field to harvest real raspberries.

environment (i.e. the raspberry plant) which is used to train the robot. For the lab-trained robot to function well in the field environment, the difference between the physical twin and its natural counter part (i.e.: the Lab2Field gap on this physical simulation), must be minimized. A key functionality of the physical twin is to provide sensory feedback of forces the raspberry experiences.

The harvesting robot comprises of a force-controlled parallel-jaw gripper with silicone fingers to increase the contact friction. The raspberry harvesting process uses two controllers: the harvesting controller, which modulates the gripping force while pulling the fruit off the plant; and the visual servoing controller, which uses on board cameras to approach and align the fingers to the raspberry. Although we consider the Lab2Field transfer in the tactile and visual domain, the development of the harvesting controller is the main focus of this work.

The development of the harvesting controller begins by obtaining the human reference response when a human harvests the physical twin. Based on the characteristics of this response, the controller can be designed and its parameters can be tuned through lab trials. Certain parameters of the controller are tuned automatically by comparing the robot and human response on a harvest. The visual servoing controller is also developed in the lab only using the physical twin setup. Once both controllers are developed, the robot is deployed in the field. The quality of the harvesting controller and the visual servoing controller is evaluated separately.

Finally, the successes and failures in the field environment are analyzed by comparing the robot performance between the lab and field setup by observing where the similarities and differences are between the physical twin and the real crop.

**Raspberry physical twin**. The goal of the physical twin is to mirror reality whilst providing force feedback. There are a large number of environmental variables which affect the harvesting

process. While it is impossible (or at least costly) to accurately model the environment to the full extent, the key properties that must be matched are those that determine success in performing and evaluating harvesting. Through observation of the fruit and human harvesting, the dominant dynamics were selected and are reflected in Fig. 2b. To represent the interaction between the harvester and the fruit, the compression force $F_c$ and pulling force $F_p$ must be considered. For the characteristic dynamics of the crop, the stiffness of the fruit on and off the plant, $k_{on}$ and $k_{off}$, the maximum pulling force required to harvest the fruit from the plant $F_{p,\max}$, and the stiffness and damping of the stem, $k_{plant}$, $d_{plant}$, are considered.

*Components of the physical twin*. The physical twin of the raspberry comprises of the fruit and the plant. The fruit was developed to specifically match the mechanical properties of the real raspberry (Fig. 2b) and sense key interaction forces shown in ref. [34]. A colored version of the sensorized fruit introduced in ref. [34] was used in this work. Among the variations given in ref. [34], the thin variant of raspberry type B was selected. The plant is a new addition to the physical twin and comprises of the receptacle (the section which directly attaches to the fruit), the stem, and the background crop. The two combine to make the physical twin setup in Fig. 2a. This simulates a single raspberry on a plant, which can be harvested and replaced repetitively by a robot without human intervention (Movie S1).

Fig. 2c shows details of the fruit of the physical twin. The fruit of the physical twin comprises of a silicone structure with silicone tubes wrapped within it to form a soft fluidic sensor. A magnet is placed both on the receptacle and the fruit, of which the separation distance can be tuned via a screw, to vary the maximum pulling force $F_{p,\max}$. The stem of the plant is fabricated from a curved 3D printed TPU flexure reproducing the compliance of the stem $k_{plant}$, $d_{plant}$. The visual appearance of the setup is designed to show resemblance of a real raspberry

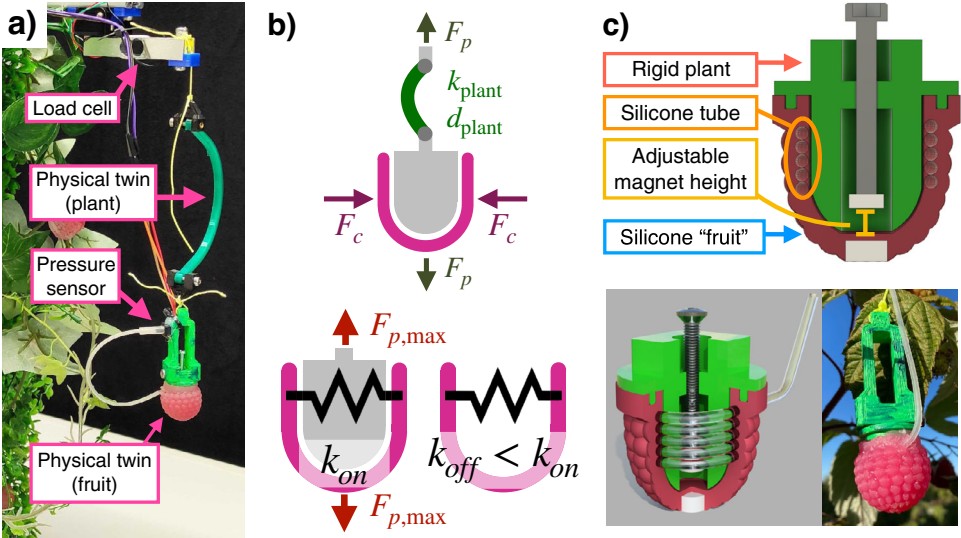

**Fig. 2 The raspberry physical twin setup and its schematic. a** The physical twin setup, comprised of the fruit and the plant. The fruit is hung from a load cell connected by a curved piece of 3D printed TPU, which represents the plant. **b** Dominant forces and characteristic dynamics which affect the harvesting of the fruit. The compression and pulling forces ($F_c$ and $F_p$) are identified as dominant forces. $F_{p,\max}$, $k_{on}$, $k_{off}$, $k_{plant}$, and $d_{plant}$ are identified as key dynamics. **c** Detailed schematic, rendering, and image of the physical twin.

crop, including the color of the fruit/plant, and the form and color of the additional foliage behind the fruit.

*Closing the reality gap.* To close the Lab2Field reality gap, the key characteristics, $k_{on}$, $k_{off}$, and $F_{p,\max}$ are tuned to match the real fruit. In comparison with 14 exemplar real raspberries, the fruit was designed such that the key characteristics lie within the bounds observed in the real raspberries. The compression force vs compression distance plots shown in Fig. S1a show the response from the physical twin lies within real raspberries (hence matching $k_{on}$ and $k_{off}$). Fig. S1b shows the pulling force not only lies within real raspberries, but can be tuned accurately within this bound. Supplementary note 1 explain in detail regarding Fig. S1.

*Sensor feedback from the physical twin.* The sensor response of the twin should relate directly to the interactions from the harvester to allow for optimization and evaluation of the harvesting process. To achieve this, the physical twin must sense the two dominant interaction forces $F_p$ and $F_c$. $F_c$ is measured through a pressure sensor embedded in the fruit, and $F_p$ is measured through a load cell above the physical twin (see Fig. 2a). The compression force is relevant to know the moderation of force needed at the various stages of the harvest, and the pulling force is relevant to identify the moment of the fruit coming off of the plan. If the physical twin is able to sense the dominant forces, the response from the sensor can be used as a metric to directly compare the quality of harvest among different controllers, robots, and/or humans. This allows for a quantitative metric to be used to improve and tune the controller.

**Design and tuning of the harvesting controller.** The realization and training of the harvesting controller is made possible through the physical twin (Fig. 1). By first acquiring a representative sample of human demonstrations, the form of the controller can be realized. Then, parameters of the harvesting controller are tuned using the physical twin. We demonstrate the robot can automatically tune parameters by repetitively harvesting the physical twin and updating the parameters iteratively.

*Human harvesting reference.* The first step of designing the controller is acquiring the human reference signal. This process, outlined in Fig. 1d, is achieved by a human harvesting the physical twin ten times by hand, and recording the measured forces $F_{c,T}$ and $F_{p,T}$. A representative signal was chosen manually out of the 10 signals (some are shown in Fig. 3b).

To handle the variety in ripeness of raspberries in the field environment, three different physical twin *settings* were considered by varying $F_{p,\max}$. Three values of $F_{p,\max}$: 1.4N, 2N, 2.8N, are used and will be referred as the Low, Medium, and High pulling force settings. Fig. 3b shows a representative sensor response of the physical twin for a human harvester for each of these pulling force settings. From these demonstrations, key features that describe a successful harvest are identified. For example, $F_p$ gives a clear indication of the fruit being harvested from the plant as the force drops to zero at moment of detachment. Post-harvesting, the compression force greatly reduces to prevent squashing of the fruit. The characteristics are similar across all three settings, while the magnitude of the forces varies.

*Form of the harvesting controller.* Based on the human demonstrations, the form of the controller and the sensing on the robot gripper can be designed. The harvesting robot is equipped with a gripper that can grasp the fruit with a parallel jaw mechanism. Fig. 3a shows the diagram for the gripper-fruit system annotated by the true and measured forces by the robot and the physical twin. The robot gripper has two load cells (one on each finger), targeted to directly measure the same quantities as the physical twin, denoted as $F_{c,R}$ and $F_{c,R}$.

The harvesting controller is designed to have three phases: A, B, and C as labeled in gray on Fig. 3c. In phase A, the gripper applies a compression force of $F_c^1$. This force is desired to be the *minimum sufficient* force to harvest the fruit. Once $F_c^1$ is reached (gray dotted line on figure), phase B starts where the robot arm moves downwards at 10 mms$^{-1}$. Phase B ends when the harvest is detected (red dotted line), by detecting a negative change in $F_{p,R}$ of some threshold $F_{p,\text{harvest}}$. In phase C, the vertical motion is stopped, and a new compression force setpoint $F_c^2$ is set. This

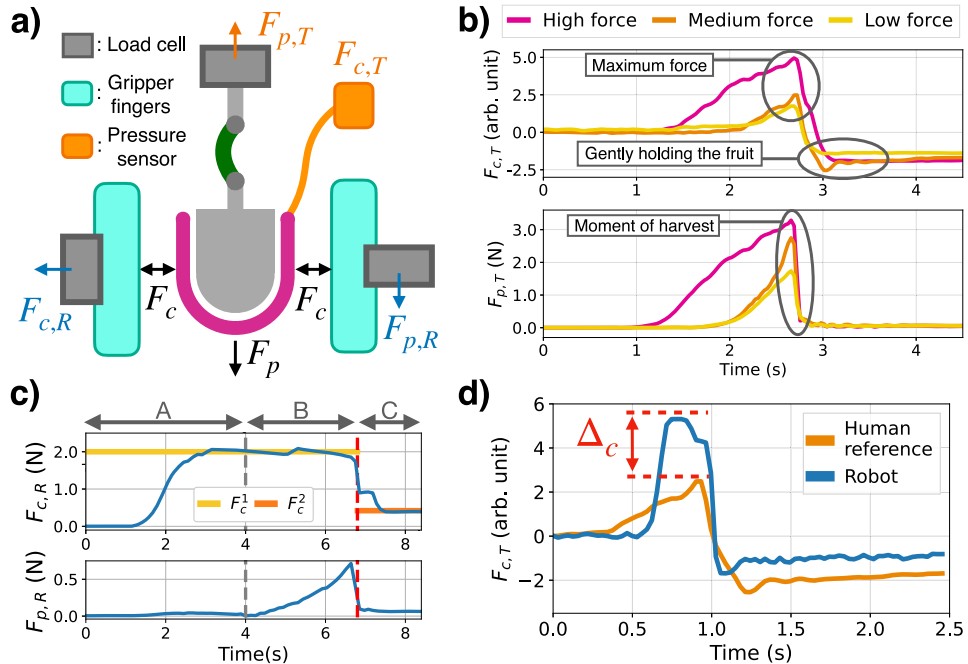

**Fig. 3 Schematic and sensor response of the robot and physical twin during the harvesting motion. a** Representation of the interactions and measurements between the physical twin and the robot gripper. Subscript $c$ and $p$ distinguishes the compression and pulling forces. Subscript $T$ and $R$ distinguishes measurements by the twin or robot. **b** Sensor response from the physical twin when harvested by a human with salient features highlighted. The physical twin is tuned to three settings corresponding to a High, Medium, and Low maximum pulling force. **c** Time series response of the robot's force sensors to describe the form of the harvesting controller divided into phases A, B, and C. **d** A comparison between the human and robot harvesting performance through the compression force measured by the physical twin. The error $\Delta_c$ is to be minimized automatically by the robot adjusting it's control parameters.

force allows to hold the raspberry gently, similar to how a human (see Fig. 3b).

*Automatic controller turning*. The control parameters are can be tuned through iterative automatic harvesting experiments with the physical twin. The block diagram of this optimization process is illustrated in Fig. 1b. Using the un-tuned controller, the robot performs the harvesting task on the physical twin. The sensor response from the physical twin is compared against the human reference and the controller parameters are updated accordingly. Of the three parameters of the harvesting controller, $F_c^1$ is tuned automatically. The other two variables are tuned manually using the physical twin. $F_c^2$ is tuned to be the lowest force setting capable of holding the physical twin without dropping it. $F_{p,\text{harvest}}$ was a threshold chosen to detect most pulls tested in the lab environment.

Since the physical twin can be repeatedly harvested, the robot is able to continuously perform this robot testing and parameter updating process by harvesting and re-attaching the fruit on the plant after each trial (Movies S1 and S2). To tune $F_c^1$, we wish to minimize the error between the maximum compression force read by the physical twin when harvested by a human and a robot. Fig. 3d shows this quantity as $\Delta_c$. This error can be used to update the force setpoint at iteration $i$, $F_{c,i}^1$, to generate the force setpoint at step $i + 1$, $F_{c,i+1}^1$, through a gradient based method detailed in section "Automatic controller tuning implementation".

The result of this tuning with the Low pulling force setting is shown in Fig. 4 (the corresponding results for the Medium and High pulling force settings are shown in Fig. S2 and explained in Supplementary note 2). Although $F_c^1$ is a single force setpoint to be tuned, through real world experiments it is able to capture the apparent variability in the robot-physical twin interaction.

Especially in the context of soft body interaction, such as this task, subtleties of the system can dictate the overall behavior.

**Lab development of the visual servoing controller.** The physical twin can also be used to design and develop the full robotic pipeline which incorporates the visual detection and motion planning to move the gripper fingers around to fruit, so it can be harvested.

This pipeline is shown in the top flowchart in Fig. 5. After detecting a target raspberry (approx. 20-40 cm away from the fruit), the gripper first moves itself horizontally and vertically so the target fruit is centered in the image frame. Then the gripper moves forwards in the direction out of the image plane to approach closer to the fruit (approx. 10 cm from the fruit). Finally, the gripper is moved slowly towards the fruit to precisely align the fruit within the gripper.

The accuracy of the visual servoing controller in the lab is shown in Fig. 6b. For both the $\Delta x$ and $\Delta y$ directions (defined in Fig. 6b), the gripper center is consistently within ± 10 mm with a mean of 2.9 mm and 3.1 mm, and a standard deviation of 2.2 mm and 2.5 mm respectively. Although the $z$ (vertical) variation is not measured, its accuracy is similar to that of $x$ as it uses the same visual servoing procedure for alignment. This result is obtained by running the detection sequence 27 times. In all attempts the alignment succeeded without any catastrophic failures.

**Field test results.** The raspberry harvesting pipeline developed in the lab is directly transferred to the field environment. Snapshots of the full harvesting pipeline in the lab and field environment is shown in Fig. 5 (see movies S2 and S3 for the full pipeline demonstrated in the lab and field respectively). For a single

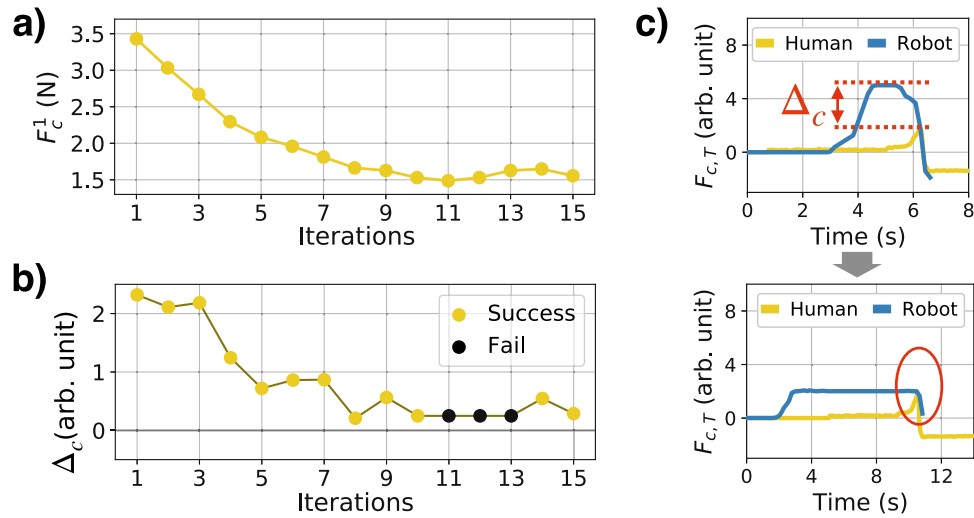

**Fig. 4 Automatic controller tuning for the low pulling force setting on the physical twin. a** The change in $F_c^1$ over the 15 iterations. **b** How $\Delta_c$ (the error to be minimized) varies over the 15 iterations with indication of a failed harvest. **c** The compression force experienced by the physical twin before and after controller tuning. Before, a large $\Delta_c$ is visible between $F_{c,T}^{Robot}$ and $F_{c,T}^{Human}$, which is minimized after the tuning.

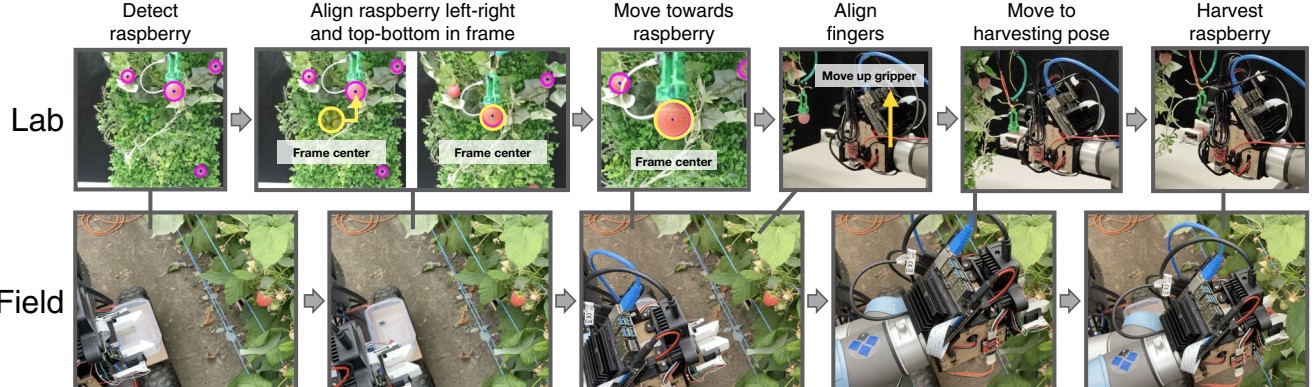

**Fig. 5 Images sequence of the full robotic pipeline of raspberry harvesting both in the lab and the field.** The top row of images correspond to the full robotic pipeline in the lab. The bottom row of images correspond to the full robotic pipeline in the field.

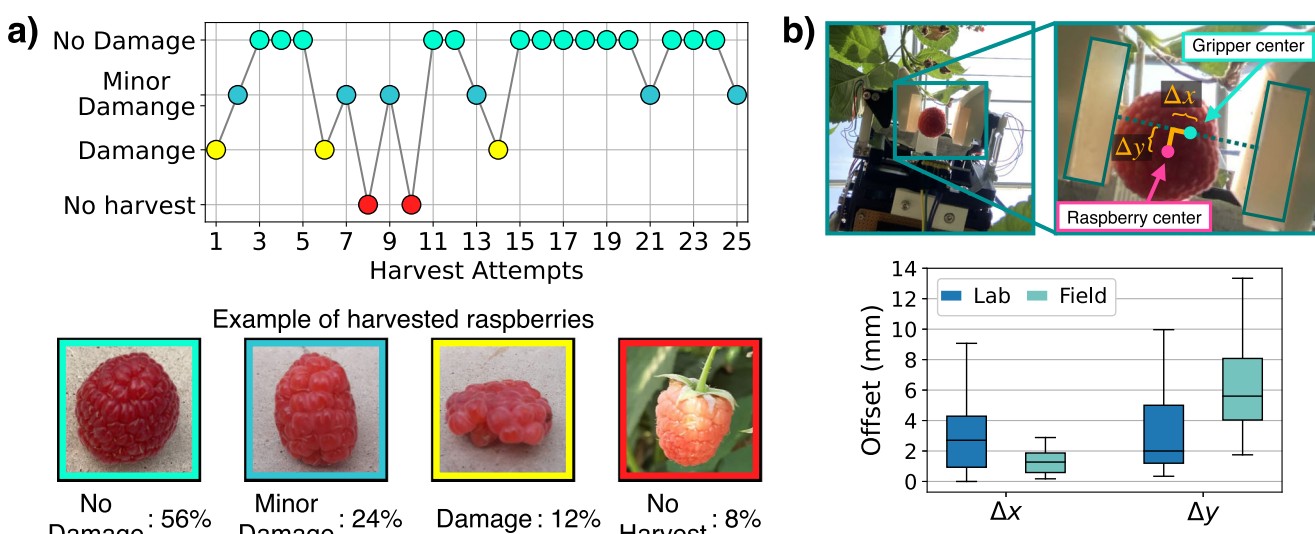

**Fig. 6 Performance of the harvesting and visual servoing controller. a** Performance of the harvesting controller over 25 harvesting attempts and images of raspberries corresponding with four result categories. **b** Performance of the visual servoing controller by measuring the offset of the fruit from the center of the gripper fingers for both lab and field conditions.

raspberry visible in the frame without occlusion, firstly the robot approaches to the harvesting position (visual servoing controller). Secondly the robot harvests the fruit (harvesting controller).

*Performance of the harvesting controller.* The harvesting controller with no further tuning or adjustments was tested on case-study of 25 raspberries visually determined to be ready for harvest, but with a range of size and ripeness. The raspberries are of the *Rubus idaeus* species and were grown in an open-air green house environment. The result of each harvest was categorized into four outcomes: No harvest, Damage, Minor Damage, and No Damage; summarized in Table S1 with corresponding explanations in Supplementary note 4. In addition to these harvesting tests, more extensive testing was performed on a larger number of plants and raspberries.

Fig. 6a summarizes the 25 exemplar harvesting experiments, recording 60% no damage and 20% minor damage harvest. Our discussions with local raspberry growers revealed that they estimated that humans had a 90% harvesting success rate with minor damage. The cause of no harvest raspberries are either due to the fruit being less ripe than anticipated, or a large damage caused to the outer surface of the fruit being scraped before being pulled off. Harvests labeled with damage were caused by a failure in the controller to detect the moment of harvest. Harvests labeled with minor damage are resultant of the deformation caused to the fruit during the transition from on the plant to off the plant.

*Performance of the visual servoing controller.* The performance of the visual servoing controller evaluated in the field is summarized in Fig. 6b. 11 approaches to three different raspberries (three approaches per fruit, starting from different initial configurations) were tested.

The offset error $\Delta y$ is larger than that of $\Delta x$ with a mean value of 1.5 mm compared to 6.1 mm. This is expected, as the forward approach to the raspberry (causing $\Delta y$) is dependent on the size

of the fruit which is variable. Whereas, the side-to-side approach (causing $\Delta x$) being theoretically invariant to the size of the fruit, is comparable in magnitude (in fact performed better) to the performance in the lab.

*Performance of the full pipeline.* Combining the harvesting and visual servoing controllers, the full robotic pipeline was tested on four untested raspberries summarized in Table 1. For each raspberry, the pipeline was tested until successfully harvested. In aggregate the pipeline was successful 4 out of 7 attempts. Three representative successful harvests are shown in Movie S3. The failure cases are further elaborated in section "Failure cases of the robotic system".

**Successes and limitation on the lab-to-field transfer.** To explore the extent to which the physical twin is essential to enabling the successful harvest, we can compare the behavior of the harvesting robot in response to the lab and field experiments. Secondly, we can explore the limitation of both. Under the concept of the physical twin, if the key characteristics are matched, we would expect successes and failures in the lab to be reflected in the field.

*Robot sensor response in the lab vs field.* The sensor measurement from the robot's fingers ($F_{c,R}$ and $F_{p,R}$) can be used to compare the performance of the robot between the lab and field. Fig. 7 show the time series data in the lab and field for the three different physical twin setting (Low, Medium, and High pulling force setting corresponds to Fig. 7a–c).

In all cases, the general shape of the sensor response in the field matches that of the lab. For the compression forces, for both lab and field, a characteristic "bump" is seen in the transition from $F_c^1$ to $F_c^2$. This is likely caused by the fruit detaching from the plant but still some area is in contact. For the pulling force, the sensor response gradually increases as the plant is pulled down, until a sudden drop when the fruit is harvested. Such similarities highlight the close matching of the physical twin and the real fruit, which explains the 80% minor-to-no damage harvest rate directly from the lab to the field transfer.

However, there are also differences in the sensor response. One difference is the variability in the pulling force increase until the moment of harvest. Unlike the physical twin, the real plant shows more variable and complex behavior when pulled. Another difference is that the maximum pulling force of the real fruit is lower than anticipated. For raspberries which required a *medium* or *high* force setting to harvest, a lower maximum pulling force compared to physical twin was recorded. This suggests that for such raspberries, either the surface friction coefficient (which was not tuned on the twin) was lower, or the geometry of that fruit

| Table 1 Result of the full pipeline testing | | | |
|---|---|---|---|
| **Tested fruit** | **Attempt 1** | **Attempt 2** | **Attempt 3** |
| Raspberry 1 | Success | - | - |
| Raspberry 2 | Fail | Fail | Success |
| | Detected fruit behind leaf | Gripper cannot reach fruit | |
| Raspberry 3 | Success | - | - |
| Raspberry 4 | Fail | Success | - |
| | Fruit beyond range of arm | | |

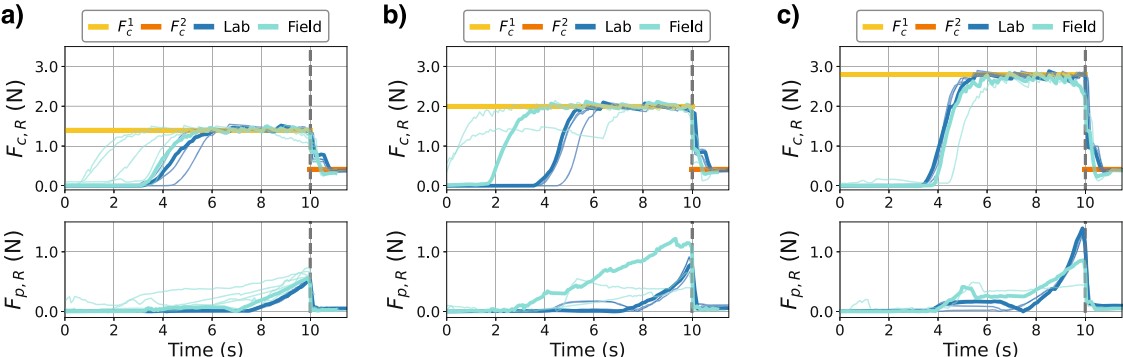

**Fig. 7 Comparison of the robot's force measurements on three pulling force settings in the lab and field.** Time series plots comparing the robot's force sensor measurement, $F_{c,R}$ and $F_{p,R}$, both in the lab and in the field. **a–c** Corresponds to the pulling force settings Low, Medium, and High respectively.

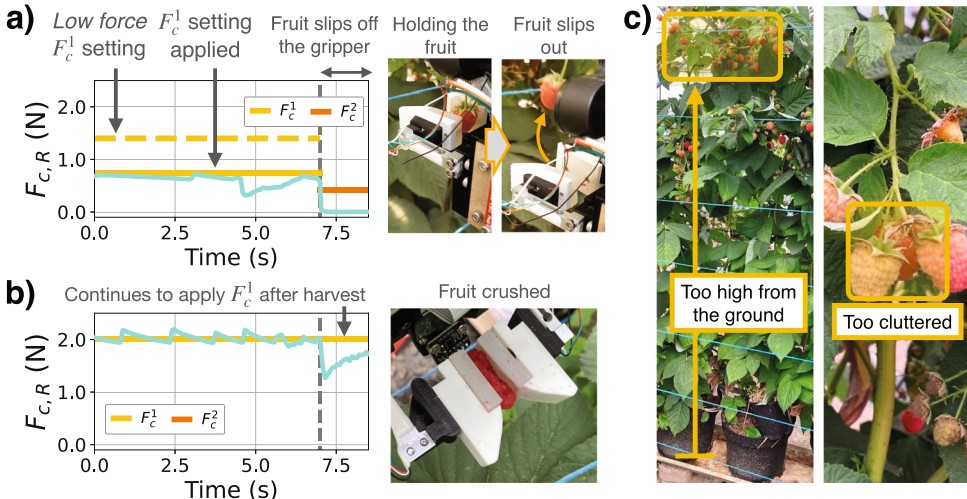

**Fig. 8 Failure cases of the developed harvesting robot. a** Failure due to the first compression force setting $F_c^1$ being too low, and the fruit slips off the gripper fingers. **b** Failure due to the controller not detecting the moment of harvest. The fruit is crushed after it is removed off the plant. **c** Failure due to the fruit being located where the gripper cannot reach.

meant only a small contact area was achieved compared to other raspberries.

*Failure cases of the robotic system.* By considering the failure cases in the field, the validly of the form of this controller, and the limitations of the physical twin can be discussed.

Fig. 8a and b show the sensor response for measuring the compression force $F_{c,R}$. In Fig. 8a, the force setpoint $F_c^1$ is deliberately lowered to be 0.8N, which is lower than the low pulling force setting. Here the robot is unable to exert the necessary force on to the raspberry, and therefore the fruit slips off the fingers. This shows the range of forces tuned through the physical twin is correct; and if the difference between the physical twin and the real fruit was further apart, the force setpoint may be too low, and would result in an unsuccessful lab-to-field transfer.

In Fig. 8b, the detection of harvest is deliberately ignored. Here the robot starts with the low force setting and continues to apply that force even after the fruit is off the plant. When the fruit detaches from the plant, the compression reaction force momentarily drops as there is a step change in stiffness on and off the plant. However, the gripper continues to "squish" the fruit until the setpoint is re-reached. The resultant fruit is completely damaged as in the photograph. This failure case validates the form of this controller designed by considering the physical twin.

Finally, in Fig. 8c, two conditions of the raspberries in the field where the robot fails to harvest are shown. In one case, the raspberry is out of reach of the range of the robot arm. In another case, the raspberry is cluttered by other raspberries which are unripe. Both of the two example cases are not captured by the physical twin, as the physical twin was a single raspberry hanging vertically in an uncluttered space in range of the arm. When these conditions are met in the field environment, the harvest is successful, but otherwise the robot fails. These two failure cases are apparent in the full pipeline test with results shown in Table 1. The two failed attempts with Raspberry 2 can be attributed to the cluttered case. The one failed attempt with Raspberry 4 is due to the arm too short to reach the fruit.

## Discussion
The concept of digital twins enabled a paradigm shift for many industries including agriculture[35,36]. We extend this concept to the physical domain with the creation of physical twins. In previous work, similar concepts have been explored for evaluating robot performance in domains including medical robotics with the development of sensorized phantoms[37,38], or in assistive robotics with the development of sensorized tools[39]. Although our work builds upon such ideas, we extend the concept to include the idea of teaching the task via human demonstration, with the physical twin providing a common currency or means of measuring and recording the physical interactions. In addition, we experimentally validate the transfer from Lab2Field, and seek to demonstrate that the success arises from the use of the physical twin.

We demonstrate that physical twin provides sufficient similarity to the real world system to allow transfer of the controller with a high success rate. It importantly shows the same failures occur in the field, demonstrating the specificity of the physical twin. The capabilities of the robot could be further extended and improved by extending the physical twin concept to mirror the larger complexities of the plant. This could include modelling bunches of raspberries, better matching the surface properties and structural properties of the raspberry, and wider variety in the range of visual situations that are represented. Furthermore, extending the simulation to include variable environmental conditions such as lighting, temperature, and humidity could further close the Lab2Field reality gap. Introducing greater variety into the physical twin beyond the pulling force stiffness could help to better represent the range of over 130 varieties of raspberries that are present around the world[40], which includes some varieties that are white so show consideration visual variation[41].

This approach of leveraging a physical twin is particularly suited for harvesting due to the costs of in-field experimentation and also the short harvesting period for crops which limits the testing and evaluation period. Within the domain of harvesting, this method has potential for crops which require delicate tactile interactions, and where there is uncertainty or variability within the crop. This could include other berries, funghi, and leafy vegetables.

By developing robotic solutions for harvesting crops such as raspberries we provide means of reducing costs, reducing the reliance on increasingly hard to find labour sources, and improve working conditions[16]. There is also the potential to explore other benefits that arise from robotic harvesting. For example, crops could be harvested early in the day, or at night, when water content of the fruit is higher, maximizing sugar content[42,43]. We could also move to precision harvesting throughout the day,

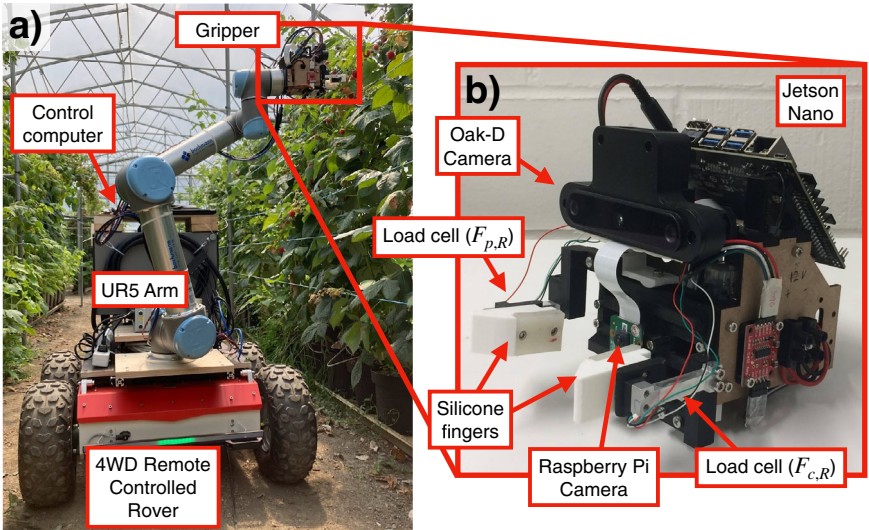

**Fig. 9 The setup of the robotic system used to harvest the raspberry. a** The full robotic setup with the mobile base, robot arm, and the gripper. **b** Detailed figure of the gripper and its functional components.

optimizing the harvest of every berry and leveraging machine learning methods to perform the necessary localization and classification at the individual fruit level[44]. This would require an extension of the current demonstrated technologies to a fully autonomous and increasingly robust system that is fully deployable in the field. In this work we focused on the final harvesting motion of the robotic system using the physical twin. In order to reach a fully functioning harvesting robot, outdoor navigation, error recovery, and improved manipulator planning must be implemented. Furthermore, to reduce the production cost of the robotic system, alternative hardware designed specifically for this task can be used. For example, a Cartesian gantry can be used instead of a 6-axis robotic arm, which is common in existing harvesting robots[45–47] including implementation of autonomous navigation within the crops, error detection and recovery, and decision marking regarding which fruit to harvest.

Extending the physical twin for more general manipulation could also be beneficial in a number of scenarios. Firstly, it allows for human training or a human reference point, however, further work is required to automate the extraction of the parameterzed controller and associated update policy. Secondly, it can be used for tasks that are challenging or computationally expensive to simulate yet easy to model in the real world. There is also scope here for the physical twin to be used to evaluate why the sim2real gap exists for certain manipulation tasks. This could be performed by comparing the interactions with a robot in a simulation and a physical twin, and also, purposefully altering aspects of the physical twin to simulate error in sim2real transfer and understanding the resulting impact on robot performance.

## Methods
**Harvesting robot**. The Raspberry harvesting robot is shown in Fig. 9 with the full robot shown in Fig. 9a alongside the detailed image of the gripper on the right (Fig. 9b). The full robot is a mobile manipulator combining a 4WD mobile base (Husarion Panther) and a 6 degree of freedom robotic arm (Universal Robotics UR5 Arm). The control computer is mounted on the rear of the mobile base where a human can operate the robot. Fig. S3 and Supplementary note 3 shows and explains the system integration diagram.

The gripper is a custom made parallel jaw gripper which is achieved through a rack and pinion mechanism actuated using a Dynamixel XM430-W210-R motor. The finger tips of the gripper is a combination of 3D printed PLA with the surface that touches the raspberry covered with casted silicone (SmoothOn DragonSkin-10). The fingertips are connected to the gripper via a uniaxial 500 g load cell on both fingers mounted in such a way that one finger read the vertical force $F_{p,R}$ and the other reads the horizontal force $F_{c,R}$. In between the fingers, a Raspberry Pi Camera is located

alongside a time-of-flight sensor to help align the raspberry. Above the fingers, a stereo-vision camera (Oak-D Camera) is mounted, also used to align the raspberry. Behind the fingers, a Jetson Nano board is placed to process the Raspberry Pi camera data. Within the gripper, a microcontroller (Arduino Nano Every) and a U2D2 module (to communicate with the Dynamixel motor) is mounted.

**Harvesting controller implementation**. The harvesting controller takes the form of standard PID controller. The input to the PID controller is the force setpoint $F_c^1$ or $F_c^2$. The motor is controlled through a velocity control input, where a PID controller located inside the servo motor regulates its velocity based on the target velocity.

For the two setpoints of the force controller, two different sets of PID gains are tuned ($k_{p,1} = 0.05$, $k_{i,1} = 8 \times 10^{-6}$, $k_{d,1} = 0$ for $F_c^1$, and $k_{p,2} = 0.3$, $k_{i,2} = 10^{-4}$, $k_{d,2} = 0$ for $F_c^2$), and the Ziegler-Nichols method was used to tune both gains. The transition between controllers does not cause any instability issues as the switching condition is irreversible.

**Visual servoing controller implementation**
*Raspberry detection*. At the core of the alignment pipeline is the raspberry detector. The detector uses classical computer vision techniques to localize and estimate the size of the raspberries through approximating them as circles.

The raw image is first thresholded in the HSV space. The value is used to threshold the brightness of the environment, and the hue is thresholded to identify the pink color of the fruit to produce a binary mask to segment pixels which contain the fruit from all other pixels. After applying a Gaussian smoothing to this mask, the Circular Hough Transform (CHT) is applied, which outputs the center location and radius of potential raspberries. By knowing the approximate size of the fruit in the frame, the localization is successful.

*Cartesian alignment*. The alignment procedure aims to align the gripper center to the fruit by moving itself in Cartesian directions (see Fig. 5a). The first step is to align a target raspberry to the center of the image frame. In this process the image from the Oak-D camera (mounted above the gripper fingers) is used. Before any movement of the gripper, the target raspberry is selected by detecting all raspberries within the frame, and choosing the one which is closest to the frame center. Once the target fruit is chosen, the UR5 arm is commanded to move first in the horizontal direction, and then in the vertical direction to align the center of the detected fruit to the center of the camera.

This alignment in the image frame is necessary for the second step, which is to move in the direction towards the fruit. With the fruit at the center of the image frame, the location of the fruit on the frame is unaffected by perspective effects. The approach to the fruit is performed in two motions. The first motion is to move close until the detected raspberry in the image frame exceeds a certain radius (40 pixels was chosen heuristically based on experiments). The second motion is to position the fruit at the center of the two gripper fingers. The fruit is assumed to be positioned correctly when either the Raspberry Pi camera or the time-of-flight(ToF) sensor, both mounted level with the fingers, detect the fruit through thresholding. While the ToF sensor directly measures the distance, the Raspberry Pi camera does this by counting the number of pixels which belong a raspberry in its frame. The segmentation is performed identically to the raspberry detection algorithm (section "Raspberry detection") before applying the CHT.

**Measuring the performance of the visual servoing.** The performance of this visual servoing is determined by measuring the offset of the fruit from the gripper center just before the moment of harvest. In the lab, the alignment sequence was run from 27 unique starting locations (where the target raspberry is in frame). The 27 unique locations are given by ±8 cm deviations in the three Cartesian directions from a point 30 cm directly away from the target physical twin.

After running the alignment sequence, a photo is taken form the bottom of the gripper to measure the relative location of the fruit from the gripper center. In the photo, the edges of the gripper fingers and the center of the fruit is identified manually (a human clicking on the features of a photo displayed on a screen). For every photo, this manual process is performed five times to minimize human error.

**Automatic controller tuning implementation.** The controller parameter $F_c^1$ is tuned automatically by the robot system (see section "Automatic controller turning"), and is fully described in Algorithm 1.

**Algorithm 1.** Automatic tuning of $F_c^1$

```
 1: Start iteration i ← 1
 2: Set initial guess of F_c^1 : F_{c,i}^1
 3: while i <= 15 do
 4:     for j ← 1 to 5 do        ▷ Repeat the fruit harvesting 5 consecutive times
 5:         harvest(F_{c,i}^1)
 6:         if Harvest is successful then
 7:             P_{i,j} ← max(F_{c,T})
 8:         end if
 9:     end for
10:     if All 5 harvests are successful then
11:         P̄_i ← mean(P_{i,1:5})
12:         Δ_c ← P̄_i − P̄^H        ▷ P̄^H is the maximum F_{c,T} recorded by a human
13:         if Δ_c > 0 then
14:             F_{c,i+1}^1 ← F_{c,i}^1 − γΔ_c        ▷ In this experiment γ ← 10
15:         else
16:             F_{c,i+1}^1 ← F_{c,i}^1 − F_rand   ▷ F_rand is a random value between 1 and 10
17:         end if
18:     else
19:         F_{c,i+1}^1 ← F_{c,i}^1 + (F_{c,success}^1 − F_{c,i}^1) n_fail/5  ▷ n_fail is the number of failed
                                                                            attempts
20:     end if
21: end while
22: return F_c^1 = F_{c,15}^1
```

Algorithm 1 takes 15 iterations (denoted by subscript $i$) to tune $F_c^1$. During this iteration, the value of $F_c^2$ and $F_{p,\text{harvest}}$ was kept constant both at 0.196N (20 gf). At every iteration, the robot will harvest the physical twin five consecutive times (denoted by subscript $j$). The repeated harvests for a single force setpoint $F_c^1$, aims to minimize the effect of stochasticity in the system. The precise contact location of the gripper on the physical twin can change the interaction dynamics. Furthermore, the sensor measurement of the soft sensor (due to the deformation of the material) can induce some drift and noise.

For every attempts within the five trials, if the harvest is successful (the fruit is off the plant and is held by the gripper with $F_c^2$ after the harvest), the maximum compression force recorded by the physical twin, $\max(F_{c,T})$, is recorded as $P_{i,j}$.

Once the five harvests are performed for $F_c^1$, $F_{c,i+1}^1$ is generated through an update rule described in lines 10-19 in Algorithm 1. The update rule has three cases.

(i)   All five harvests are successful and the robot is **worse** than the human ($\Delta_c > 0$)
(ii)  All five harvests are successful and the robot is **better** than the human ($\Delta_c \leq 0$)
(iii) Not all harvests are successful

In case 1, the update rule is described in line 14 of Algorithm 1, and is derived from a simple gradient decent based rule shown in equation (1), which aims to tune $F_c^1$ to minimize the squared error $\Delta_c^2$.

We wish to tune $F_c^1$ by minimizing the error $\Delta_c^2$ : $F_{c,i+1}^1 = F_{c,i}^1 - \gamma \frac{d\Delta_c^2}{dF_{c,i}^1}$

$$\frac{d\Delta_c^2}{dF_{c,i}^1} = 2\Delta_c \frac{d}{dF_{c,i}^1}\left(\overline{P}_i - \overline{P}^H\right) = 2\Delta_c \frac{d}{dF_{c,i}^1}\overline{P}_i$$

Assuming the gripper can accurately exert its demand force such that $F_c \approx F_{c,i}^1$

and the physical measurement is proportional to the true force $F_{c,T} = \alpha F_c$,

$$\frac{d}{dF_{c,i}^1}\overline{P}_i = \frac{d}{dF_{c,i}^1}\text{mean}(\max(F_{c,T})) \approx \alpha$$

Hence, $\frac{d\Delta_c^2}{dF_{c,i}^1} \approx 2\alpha\Delta_c$ and thus, $F_{c,i+1}^1 = F_{c,i}^1 - \gamma\Delta_c$

(1)

In case 2, the robot is "better", than the human recording a negative error. In this case, rather than penalizing the robot to match the human, the force setpoint is decreased by a small random amount $F_\text{rand}$ to push the capabilities of the robot

even more (line 16 of Algorithm 1). In practice this case did not occur, which highlights the incredible performance of a human to interact with soft objects intuitively.

In case 3, the force setpoint is too low, and must be increased to a value which it can reliably harvest the physical twin. As shown in line 19 of Algorithm 1, $F_{c,i+1}^1$ is set to a value in between the current setpoint $F_{c,i}^1$ and a most recent setpoint which led to all five harvests successful, $F_{c,\text{success}}^1$. How much the setpoint is reverted to $F_{c,\text{success}}^1$ is regulated by a ratio of failed attempts, $\frac{n_\text{fail}}{5}$.

### Field test procedure

*Testing the visual servoing controller.* The visual servoing controller was tested a total of 11 times on three different un-harvested raspberries. The HSV values used for the raspberry detection was adjusted to the field conditions before the experiment. Every alignment attempt, the gripper was moved to a different starting position. The measurement procedure is identical to that in the lab, where a photo is taken from underneath the gripper fingers and relative location of the fruit is identified manually.

*Testing the harvesting controller.* The harvesting controller is tested on 25 raspberries which were isolated from other fruits, leaves, and plant segments. In this experiment, the gripper center was moved to the harvesting position by hand using the teach-pad of the UR5 robot arm. For every attempt, the Low force $F_c^1$ setting is first used to harvest the fruit. If this force is too low for the harvest, and the fruit is not apparently damaged, the harvest procedure is tested again with the Medium force $F_c^1$ setting (if this fails, the High force $F_c^1$ setting).

*Testing the full pipeline.* The full pipeline (combining the visual servoing controller and the harvesting controller) was tested on four untested raspberries. For each raspberry the robot started from a position unreachable by the robot arm. Then the mobile base was driven manually until the arm was approximately in reach of the fruit. Finally, the visual servoing controller was run, followed by the harvesting controller without human intervention (see Movie S3). For every raspberry, if a failure was detected (either automatically or manually), the robot was reset and the procedure was restarted.

### Data availability
Raw data in the form of sensor measurements are freely available through this link https://gitlab.epfl.ch/create-lab/agricultural-robotics/raspberry/raspberry-grasping. All other raw data (e.g.: images and videos) and processed data is available upon contact to the corresponding author.

### Code availability
All code used in this project is freely available through this link https://gitlab.epfl.ch/create-lab/agricultural-robotics/raspberry/raspberry-grasping. Please contact the corresponding author to request further information about the code.

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

## Acknowledgements

We would like to thank Dr. Christoph Carlen, Dr. Camilo Chiang, and employees at the Agroscope Research Center in Conthey, Switzerland for providing us with the time, assistance, and location to test our robot at a real raspberry crop.

## Author contributions

K.J. conceived the study, developed the robotic hardware, conducted field experiments, and wrote the manuscript. C.P. developed the robotic software and conducted lab and field experiments. J.H. conceived the study, conducted field experiments, and wrote the manuscript.

## Competing interests

The authors declare no competing interests.
