## [Peer Review File · Communications Engineering]

Lab2Field transfer of a robotic raspberry harvester enabled by a soft sensorized physical twinReviewers' comments:

Reviewer #1 (Remarks to the Author):

This is a good research manuscript but has zero application to a real-world situation. It is very obvious that the presented technology is not a solution for Raspberry harvesting. However, the authors have provided good documentation of their study, which I believe can be used as motivation for future research. The methodology has some novelty and the manuscript is of interest to the robotic harvesting community. I would suggest considering it for publication only after a major revision of the figures and text. Below are some suggestions for improvement:

- Title: The title is too long. Please consider shortening
- Abstract: The abstract begins with a valid problem statement, followed by objective statements that can be slightly improved for better clarification. The research hypothesis has been clearly highlighted, and the authors have concluded that their method has achieved 80% “successful” harvesting “success”. Did you mean harvesting rate?
- Line 9: That is not a valid statement, and cannot be supported with an outdated reference. That reference is almost 10 years old, from 2014. There are some robotic harvesting platforms that have been commercialized for strawberries or are being used efficiently for apple harvesting.
- Line 11-12: Which crops are you referring to? This is not true for all crops. For example, in the automated harvesting of citrus with canopy shakers, human workers are not involved.
- Line 22-23: these “challenges” could describe a number of manipulation “challenges”. One distinct “challenge”...Please revise to avoid multiple uses of “challenge”.
- Fig1, Fig2, Fig4, Fig5, are overcrowded, resulting a lengthy caption. This is not a common practice. Some of the images in the figures that are presented in the result section are in fact a part of the methodology.
- In the discussion section, the authors could have included the possibility of using arrays of linear robotic arms, similar to those that are used in apple harvesting. The fact is that one single UR arm will never be fast enough to come close to a manual harvesting rate. And if you use multiple UR arms, the investment cost cannot be justified by the farmer.

Reviewer #2 (Remarks to the Author):

This paper proposes the utilization of a sensorized physical simulator of a real raspberry plant (i.e. physical twin), to design and optimize a controller for the successful raspberry robotic harvesting. The goal is to facilitate the development of the harvesting control scheme via human demonstrations and lab experiments on the physical twin; since this twin is always available regardless of harvesting seasons, the robot can be fully trained in lab through consecutive experimentation and be ready for immediate deployment in real field, without additional control tuning. In addition to the harvesting control, a visual servoing controller is discussed as well, for the successful target crop reaching by the robot's gripper; the two controllers are integrated into one robotic system (i.e. full robotic pipeline) for raspberry harvesting.

The paper has a good structure and can be accepted for publication with minor revisions. Introduction section discusses the autonomous agriculture/robotic harvesting topic sufficiently with enough references. The paper's goal is expressed clearly and the proposed process for optimizing the harvesting controller, via utilization of the developed physical twin, is presented adequately. Results (both figures and supplementary videos) support the procedure's success. Overall, I believe the presented idea is really interesting, as is its possible application in different crops to alleviate the robotic harvesters existing problem of limited experimentation time in real field (i.e. only during harvesting season).

A few minor comments are given below.

a) Regarding the physical twin, in line 111 it is stated that "a preliminary design was shown in [30]" (with [30] the authors previous work). Are there any differences between that preliminary design and the one currently utilized in this paper? If so, any changes or possible improvements that have been done in the physical twin's design could be mentioned here.

b) In my opinion, subsection 4.5 "Field test procedure" can benefit from showing some additional results, regarding the whole integrated system's operation in real field. I think including figures from the field experiments and/or making an in-text reference to the supplementary video S3 "Full robotic pipeline – in Field" (which I assume will be published as well, in case of the paper's acceptance), will complete the system's presentation, presenting its performance as a whole.

c) Few found typos: in Fig. 2 caption "The physical twin of the fruit is hung a load cell connected", in line 338 "The motor is controller through a velocity control input", in Supplementary Material section "Haresting result categories" (in subsection title).

Answers to Reviewers

Dear Reviewers

Many thanks for your feedback. We found the suggestions relevant, and they helped to improve the paper. Please find here a detailed answer to all of your points. All edited text on the main manuscript are colored in red.

Reviewer 1

1.1.

This is a good research manuscript but has zero application to a real-world situation. It is very obvious that the presented technology is not a solution for Raspberry harvesting. However, the authors have provided good documentation of their study, which I believe can be used as motivation for future research. The methodology has some novelty and the manuscript is of interest to the robotic harvesting community. I would suggest considering it for publication only after a major revision of the figures and text.

We would like to thank the Reviewer for the accurate review, and for the insightful suggestions on how to improve the quality of the paper. We have implemented all the comments mentioned with a focus on improving the introduction with updated figures and clearer writing. Furthermore, we edited the figures within the constraints given by the formatting guidelines.

1.2.

Title: The title is too long. Please consider shortening

Thank you for this comment. We revised the title to be more concise to be:

“Lab2Field Transfer for Robotic Raspberry Harvesting Enabled by a Soft Sensorized Physical Twin”

With this change, the length of the title is also compliant with this journal’s guidelines of being 15 words or fewer.

1.3.

Abstract: The abstract begins with a valid problem statement, followed by objective statements that can be slightly improved for better clarification. The research hypothesis has been clearly highlighted, and the authors have concluded that their method has achieved 80% “successful” harvesting “success”. Did you mean harvesting rate?

Thank you for this comment. This is indeed a mistake in the text. We have edited to text, and now it reads:

“... 80% harvesting success rate was achieved ...”

The abstract also underwent major edits to comply with the journal’s guidelines of being 150 words or fewer.

1.4.

Line 9: That is not a valid statement, and cannot be supported with an outdated reference. That reference is almost 10 years old, from 2014. There are some robotic harvesting platforms that have been commercialized for strawberries or are being used efficiently for apple harvesting.

We thank the Reviewer to point this out. Indeed the landscape of harvesting robotics has moved on since this reference. To reflect this, we have modified the text to reflect the instances of commercialization and more up-to-date reviews. We also added a sentence to highlight the success of robotic strawberry harvesting, as mentioned by the Reviewer. However, it is also true, that harvesting robotics is not yet a mature field, and hence we have kept the message of this portion of the paragraph. With the changes the text is as such:

“In the last decade, harvesting robots have seen notable developments driven by improvements in the underlying enabling technologies. For instance, reports of commercialized harvesting robots have increased from very few to none [5] to approximately 20 or so cases [6, 7]. In particular strawberry picking have seen multiple commercialization success through the use of suction, compressed air blowing, and stem cutting methods [7]. However, more generally, despite over 30 years of research, harvesting robots have shown limited performance improvement [8]. Compared to humans their speed is low, the cost of each device is high, and the enabling technologies are not yet mature [6, 8]”

1.5.

Line 11-12: Which crops are you referring to? This is not true for all crops. For example, in the automated harvesting of citrus with canopy shakers, human workers are not involved.

Thank you for the comment, and indeed there are examples where human harvesters are not involved. Combined with the edits made in response to the previous comment (regarding the out-of-date reference), we removed this sentence, but instead added a new sentence highlighting the issues of sustaining the agricultural demand. The new text is as such:

“Through the growing world population [9], alongside the challenges in sustaining the agricultural work force [10](made increasingly apparent through events such as the COVID-19 pandemic[11] and Brexit[12]), there is a real need to develop new methods for harvesting research, to accelerate the development of robotic solutions.”

1.6.

Line 22-23: these “challenges” could describe a number of manipulation “challenges”. One distinct “challenge”... Please revise to avoid multiple uses of “challenge”.

Thank you for this comment, the frequent use of “challenges” does introduce confusion. Upon re-reading this paragraph, there were other confusing sentences and order of logic. Hence, we rewrote the full paragraph which now reads as such:

“The challenges in developing agricultural robots can be divided in two levels. Firstly the implementation of different robotic technologies poses a practical challenge. Integration of grippers, tactile sensors, navigation, visual localization and classification, and more, which must operate robustly requires time and expertise to achieve [16]. Secondly, the environmental conditions linked to agricultural settings poses large challenges. Outdoor environments can be variable, uncertain, and harsh. This is coupled with every crop, breed and specific instance also being subject to variability [17]. One aspect of the agricultural setting which magnifies the aforementioned challenges is the need for field tests for development and evaluation. This is extremely limiting. Crops are only ripe and ready for harvest for a very short period of time and each harvesting experiment is impossible to re-run or repeat as the specific plant and conditions are constantly changing. To accelerate the design, development and evaluation of harvesting robots, we need to remove the reliance on inefficient and costly field trials and leverage alternative methodologies to meet the escalating food needs of our growing population.”

We hope this change improves the clarity of the message of this paragraph, which is convey the challenges in developing agricultural robotics, and especially the addressing the practical challenge of trying field tests.

1.7.

Fig1, Fig2, Fig4, Fig5, are overcrowded, resulting a lengthy caption. This is not a common practice. Some of the images in the figures that are presented in the result section are in fact a part of the methodology.

Thank you for this feedback. We agree the figures mentioned have multiple sub-figures and their captions are long compared to many research articles in the robotics field. In this article, we have chosen to follow the formatting style by Nature Communications Engineering (<https://www.nature.com/commseng/submit/content-types#article>).

Under this guideline, figures are designed to have multiple sub-figures and their captions must be detailed so the figure can be understood without reference to the main text. This is the reasoning behind large figures and long descriptions, which is comparable when viewing other articles published in this journal. However, the Reviewer's point is valid and thus we have split large figures into two figures as well as moving figures in and out of the Supplementary Materials section to improve clarity in the main text (with the constraint of 10 display items in the main text given by the formatting guidelines). The changes are as follows:

- (i) Added titles to each figure in bold to show a clear message behind each figure.*
- (ii) Moved (what used to be) Fig. 2d,e into the Supplementary Materials to simplify this figure.*
- (iii) Added a section of Fig. S3 in the main text as Fig. 4 to bring emphasis to the automatic controller tuning.*
- (iv) Split (what used to be) Fig. 5 into Fig. 5 and Fig. 6 to simplify the figures.*
- (v) Added Table 1 from the Supplementary Materials to highlight the result from the full pipeline testing into the main text.*
- (vi) Split (what used to be) Fig. 7 into Fig. 7 and Fig. 8 to simplify the figures.*

With regard to the second part of the feedback, we agree that some elements from early on in the results section could be considered part of the methodology. However, we decided to keep them in the results for two reasons. Firstly, the formatting of the structure in this journal means the Introduction is followed right after by Results. We tried to present a small amount of methodological information prior to presenting the results so the results are understandable. Secondly, we believe the main contribution of this article is procedure of using of physical twins to develop a robot in lab, which we demonstrate will be successful in the field too. Hence, we believe the physical twin alongside the procedure of developing the controller is related to the results.

1.8.

In the discussion section, the authors could have included the possibility of using arrays of linear robotic arms, similar to those that are used in apple harvesting. The fact is that one single UR arm will never be fast enough to come close to a manual harvesting rate. And if you use multiple UR arms, the investment cost cannot be justified by the farmer.

Thank you for this comment. In this work, since the focus was the lab2field transfer of the harvesting system only, there was limited consideration towards cost reduction in the context of the feasibility of transferring this technology into the market. This is why we used a UR5 arm and an existing mobile base which was readily available in our laboratory. However, it is also true that a UR5 arm would not be practical as a harvesting robot setup, especially when a lot of the motion can be realized using a Cartesian rig (as mentioned in the comment). We modified the second to last paragraph in section 3 Discussion, as such:

“In this work we focused on the final harvesting motion of the robotic system using the physical twin. In order to reach a fully functioning harvesting robot, outdoor navigation, error recovery, and improved manipulator planning must be implemented. Although methods and technologies for such functionalities exist, they are still preliminary and incur large development costs for commercialization. Furthermore, to reduce the production cost of the robotic system, alternative hardware designed specifically tailored for this task can be used. For example, a Cartesian gantry can be used instead of a 6-axis robotic arm, which is common in existing harvesting robots [45–47]”

In this paragraph we also highlight and acknowledge the practical challenges of developing such a harvesting robot, as the cost of the robot would not only come from the hardware itself, but also from the time and expertise needed to integrate the robotic system which is still preliminary in nature.

Reviewer 2

2.1.

The paper has a good structure and can be accepted for publication with minor revisions. Introduction section discusses the autonomous agriculture/robotic harvesting topic sufficiently with enough references. The paper’s goal is expressed clearly and the proposed process for optimizing the harvesting controller, via utilization of the developed physical twin, is presented adequately. Results (both figures and supplementary videos) support the procedure’s success. Overall, I believe the presented idea is really interesting, as is its possible application in different crops to alleviate the robotic harvesters existing problem of limited experimentation time in real field (i.e. only during harvesting season).

We thank the Reviewer for the the feedback on our paper. We are glad you found the work interesting and relevant. The Reviewer’s suggestions helped identify points that needed clarification and helped improve the paper.

2.2.

Regarding the physical twin, in line 111 it is stated that “a preliminary design was shown in [30]” (with [30] the authors previous work). Are there any differences between that preliminary design and the one currently utilized in this paper? If so, any changes or possible improvements that have been done in the physical twin’s design could be mentioned here.

Thank you for this comment. The fruit is mostly similar to the one introduced in [34], with the addition of color. The addition of the “plant” in the physical twin setup is new: the compliant stem, the background foliage, and a general test setup for the robot to harvest the twin repetitively. This is now highlighted in the main text as such:

“The fruit was developed to specifically match the mechanical properties of the real raspberry (Fig. 2B) and sense key interaction forces shown in [34]. A colored version of the sensorized fruit introduced in [34] was used in this work. Among the variations given in [34], the thin variant of raspberry type B was selected. The plant is a new addition to the physical twin and comprises of the receptacle (the section which directly attaches to the fruit), the stem, and the background crop. The two combine to make the physical twin setup in Fig. 2A.”

2.3.

In my opinion, subsection 4.5 “Field test procedure” can benefit from showing some additional results, regarding the whole integrated system’s operation in real field. I think including figures from the field experiments and/or making an in-text reference to the supplementary video S3 “Full robotic pipeline – in Field” (which I assume will be published as well, in case of the paper’s acceptance), will complete the system’s presentation, presenting its performance as a whole.

Thank you for the comment. We did not realise that video S3 was not referenced in the manuscript, and more importantly the results of the full pipeline to be more clearly highlighted. To add this results and reference in the text, the following changes were made.

- (i) Referred to the video in subsection 2.4 “Field test results” as the figure contains images of the full pipeline.

“Snapshots of the full harvesting pipeline in the lab and field environment is shown in Fig. 5A (see videos S2 and S3 for the full pipeline demonstrated in the lab and field respectively).”

- (ii) Added subsection 2.4.3 “Performance of the full pipeline” which reports the result of the full pipeline testing.

“Combining the harvesting and visual servoing controllers, the full robotic pipeline was tested on four untested raspberries summarized in Table S2. For each raspberry, the pipeline was tested until successfully harvested. In aggregate the pipeline was successful 4 out of 7 attempts. Three representative successful harvests are shown in Video S3. The failure cases are further elaborated in section 2.5.2”

- (iii) Included the discussion of failure cases in section 2.5.2 “Failure cases of the robotic system”.

“These two failure cases are apparent in the full pipeline test with results shown in Table S2. The two failed attempts with Raspberry 2 can be attributed to the cluttered case. The one failed attempt with Raspberry 4 is due to the arm too short to reach the fruit.”

- (iv) Added Table S2 and the corresponding description to summarize the results of the full pipeline testing.

- (v) Added subsection 4.5.3 “Testing the full pipeline” to explain how the full pipeline was tested in the field.

“The full pipeline (combining the visual servoing controller and the harvesting controller) was tested on four untested raspberries. For each raspberry the robot started from a position unreachable by the robot arm. Then the mobile base was driven manually until the arm was approximately in reach of the fruit. Finally, the visual servoing controller was run, followed by the harvesting controller without human intervention (see Video S3). For every raspberry, if a failure was detected (either automatically or manually), the robot was reset and the procedure was restarted.”

2.4.

Few found typos: in Fig. 2 caption “The physical twin of the fruit is hung a load cell connected”, in line 338 “The motor is controller through a velocity control input”, in Supplementary Material section “Harvesting result categories” (in subsection title).

Thank you for pointing out these errors. We have fixed all of them.

REVIEWERS' COMMENTS:

Reviewer #1 (Remarks to the Author):

The authors have revised the manuscript and addressed all comments. The manuscript can be published after an editorial check.

Reviewer #2 (Remarks to the Author):

I believe that the points raised in the previous round of review have been satisfactorily addressed, with the authors answering adequately the reviewers comments. Thus, the paper can be accepted for publication.

Few typos, found in the revised manuscript, are given below:

in abstract: "The sensors on the twin allows for direct comparison..." and "...with tunable properties, used to developed the robot..."

in figure 7 caption: "...force sensor measurements in the lab and field the three pulling force settings."

in line 320: "In order to reach a fully functioning harvesting robots..."

in line 322: "...alternative hardware designed specifically tailored for this task can..."